

# Clone wars: asexual reproduction dominates in the invasive range of *Tubastraea* spp. (Anthozoa: Scleractinia) in the South-Atlantic Ocean

Katia Cristina Cruz Capel[1,2,3], Robert J. Toonen[2], Caio T.C.C. Rachid[4], Joel C. Creed[3,5], Marcelo V. Kitahara[3,6,7], Zac Forsman[2] and Carla Zilberberg[1,3]

[1] Departamento de Zoologia, Universidade Federal do Rio de Janeiro, Rio de Janeiro, Brazil
[2] School of Ocean & Earth Science & Technology, Hawai'i Institute of Marine Biology, University of Hawai'i at Manoa, Kane'ohe, Hawai'i, United States of America
[3] Coral-Sol Research, Technological Development and Innovation Network, Brazil
[4] Instituto de Microbiologia Paulo de Góes, Universidade Federal do Rio de Janeiro, Rio de Janeiro, Brazil
[5] Departamento de Ecologia, Universidade do Estado do Rio de Janeiro, Rio de Janeiro, Brazil
[6] Departamento de Ciências do Mar, Universidade Federal de São Paulo, Santos, Brazil
[7] Centro de Biologia Marinha, Universidade de São Paulo, São Sebastião, Brazil

## ABSTRACT

Although the invasive azooxanthellate corals *Tubastraea coccinea* and *T. tagusensis* are spreading quickly and outcompeting native species in the Atlantic Ocean, there is little information regarding the genetic structure and path of introduction for these species. Here we present the first data on genetic diversity and clonal structure from these two species using a new set of microsatellite markers. High proportions of clones were observed, indicating that asexual reproduction has a major role in the local population dynamics and, therefore, represents one of the main reasons for the invasion success. Although no significant population structure was found, results suggest the occurrence of multiple invasions for *T. coccinea* and also that both species are being transported along the coast by vectors such as oil platforms and monobouys, spreading these invasive species. In addition to the description of novel microsatellite markers, this study sheds new light into the invasive process of *Tubastraea*.

Corresponding author
Katia Cristina Cruz Capel,
katiacapel7@gmail.com

# INTRODUCTION

The marine environment is continuously subjected to multiple stressors, many of which are associated with human activities (e.g., over-exploitation of resources, pollution, climate change and invasive species) (*Halpern et al., 2014*; *Gallardo et al., 2016*). Among these stressors, invasive species are considered to be a major threat to biodiversity (*Molnar et al., 2008*) with the potential to quickly trigger changes in native communities and the ecosystem services and functions, which can have wide-ranging negative impacts. There are numerous examples of marine invasions which impact humans or native biota, such as in the Mediterranean Sea with the invasion of the ctenophore *Mnemiopsis leidyi*, which caused

the collapse of the fishing industry (*Shiganova, 1998*), the algae *Womersleyella setacea*, that negatively affected sponge reproduction (*Caralt & Cebrian, 2013*) and the lionfish *Pterois spp.*, responsible for a reduction in the native fish recruitment in the Atlantic (*Albins & Hixon, 2008*).

Scleractinian corals are known to play a key role in the marine environment by building structurally complex and highly diverse ecosystems (*Reaka-Kudla, 1997*). As ecosystem engineers that are under threat globally (*Hoegh-Guldberg, 1999*; *Pandolfi et al., 2003*), scleractinian corals are rarely seen as an environmental risk. However, three scleractinian species from the genus *Tubastraea* were introduced and are spreading rapidly throughout the Western Atlantic Ocean (*De Paula & Creed, 2004*; *Fenner, 2001*; *Fenner & Banks, 2004*; *Sammarco, Atchison & Boland, 2004*; *Sammarco, Porter & Cairns, 2010*; *Capel, 2012*; *Sampaio et al., 2012*; *Costa et al., 2014*; *Silva et al., 2014*), threatening native and endemic species (*Mantellato et al., 2011*; *Santos, Ribeiro & Creed, 2013*; *Creed, 2006*) and fouling man-made structures and vessels.

*Tubastraea* is an azooxanthellate dendrophyllid genus from the Pacific and Indian Oceans that was first reported in the Caribbean in 1943 (*Vaughan & Wells, 1943*). Since then, three species have been identified in the Western Atlantic Ocean: (1) *T. coccinea*, now reported along 9,000 km of coastline of the Western Atlantic Ocean from Florida (26°47′N, 80°02′W) (*Fenner & Banks, 2004*) to Southern Brazil (27°17′S, 48°22′W) (*Capel, 2012*); (2) *T. tagusensis*, along the Brazilian coast (*De Paula & Creed, 2004*); and (3) *T. micranthus* in the Gulf of Mexico (*Sammarco, Porter & Cairns, 2010*). All three are considered opportunistic species most likely associated with transport on ships and/or oil platforms in the Caribbean, Gulf of Mexico and Brazilian coast (*Cairns, 2000*; *Castro & Pires, 2001*; *Sammarco, Porter & Cairns, 2010*).

Once established, invasive species can alter the structure of local communities, displacing and outcompeting native species (*Vitousek, 1990*; *Mooney & Cleland, 2001*; *Lages, Fleury & Menegola, 2011*; *Cure et al., 2012*; *Santos, Ribeiro & Creed, 2013*; *Miranda, Cruz & Barros, 2016*). In contrast to the native range, where *Tubastraea* is largely restricted to shaded or marginal habitats, studies on oil rigs in the Gulf of Mexico have shown that both *T. coccinea* and *T. micranthus* are excellent competitors and can overgrow other species (*Hennessey & Sammarco, 2014*; *Sammarco et al., 2015*). Similarly, in Brazil, *T. coccinea* and *T. tagusensis* can cover up to 100% of the available surface in some areas (*Mantellato et al., 2011*), killing native and endemic coral species upon direct contact (*Creed, 2006*; *Santos, Ribeiro & Creed, 2013*; *Mantellato & Creed, 2014*; *Miranda, Cruz & Barros, 2016*).

Fast growth rate, rapid range expansion, early reproductive age, propagule pressure and a wide variety of reproductive and survival strategies are biological characteristics usually associated with invasion success (*Sax & Brown, 2000*; *Sakai et al., 2001*; *Lockwood, Cassey & Blackburn, 2005*; *Sax et al., 2007*). *Tubastraea* species possess all of these characteristics (*Cairns, 1991*; *Ayre & Resing, 1986*; *Glynn et al., 2008*; *Harrison, 2011*; *Capel et al., 2014*; *De Paula, Pires & Creed, 2014*), which are enhanced by the fact that within the invaded areas they generally lack natural predators and dominant competitors. In addition, a large number of infested vectors (e.g., oil platforms and monobuoys) have been recorded

transporting *Tubastraea* spp. along the Brazilian coast, leading to rapid range expansion throughout the Southwestern Atlantic Ocean (*Creed et al., 2016*).

Asexual reproduction improves coral ability to reach high abundance (*Ayre & Miller, 2004*) and may be an important trait of many invasive species, mainly in the first stage of invasion (*Taylor & Hastings, 2005*). When associated with early reproductive age and high propagule pressure it can rapidly increase abundance. Asexual production of brooded planulae has been reported in several anthozoans, including actinarians (*Ottaway & Kirby, 1975*; *Black & Johnson, 1979*), octocorals (*Brazeau & Lasker, 1989*) and scleractinians (*Stoddart, 1983*; *Ayre & Resing, 1986*). Although *T. coccinea* and *T. diaphana* appear to reproduce mainly by asexually produced larvae (*Ayre & Resing, 1986*), there is no information for their congeners, and the proportion of sexual *versus* asexual reproduction remains unknown within the genus. Furthermore, *Ayre & Resing (1986)* were able to score only two allozyme loci to infer asexual production of brooded larvae of *Tubastraea* spp. and the use of a larger number of more polymorphic loci, such as microsatellites, is desirable to corroborate their findings.

Although *Tubastraea* species are spreading rapidly and changing local benthic communities throughout the tropical Western Atlantic, information about their genetic diversity and reproductive strategies are still scarce. The study of reproductive strategies of invasive species is fundamental to understanding the invasion process, preventing new invasions, development of effective management strategies, and resolving the ecological and evolutionary processes involved in their invasion success (*Sakai et al., 2001*; *Sax et al., 2007*). However, to date there was no molecular marker developed to perform such studies with *Tubastraea*. Here, we report 12 novel microsatellite loci specifically developed for *T. coccinea* and cross-amplified in *T. tagusensis* and investigate the clonal structure and genetic diversity of populations of these alien invasive corals in the Southwestern Atlantic Ocean.

## MATERIALS AND METHODS

### Sampling and DNA extraction

Microsatellite development was performed using samples of *T. coccinea* collected from Búzios Island (23°47′S, 45°08′W, 6 m in depth) and also from a monobuoy (IMODCO 4) at the São Sebastião channel (23°48′S, 45°24′W, 5 m of depth), Brazil. Additional samples of *T. coccinea* and *T. tagusensis*, collected from Todos-os-Santos Bay (TSB), northeastern Brazil (12°49′S, 38°46′W), and Ilha Grande Bay (IGB) (23°06′S, 44°15′ W), southeastern Brazil (∼24 colonies/species/locality), were used to test the markers and evaluate their genetic diversity (Fig. 1). Samples were preserved in 96% ethanol or CHAOS buffer (*Fukami et al., 2004*) prior to extraction. Total DNA was extracted using the Qiagen DNeasy tissue and blood kit following the manufacturer's instructions or using the Phenol:Chloroform method described by *Fukami et al. (2004)*.

### Microsatellite development and primer testing

Two genomic libraries were constructed at the National Laboratory for Scientific Computing (LNCC, Petrópolis, Brazil) using the 454 Genome Sequencer FLX platform

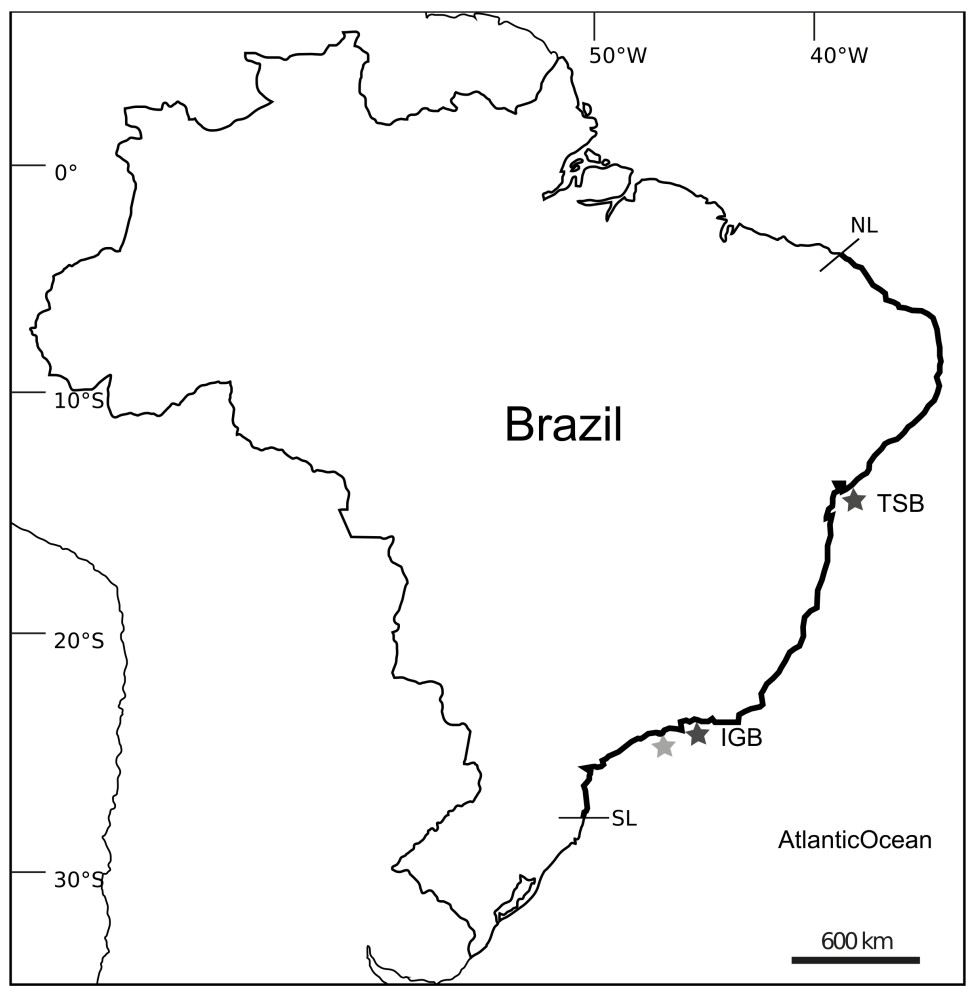

**Figure 1** **Distributional range and sample localities on Southwestern Atlantic.** Map showing the distributional range of *Tubastraea* spp. on Southwestern Atlantic with the northern (NL) and southern (SL) limits of the distribution and sampled localities: Todos-os-Santos Bay (TSB) and Ilha Grande Bay (IGB) are showed by dark-gray stars; light-gray star represent Búzios Island and São Sebastião channel where initial collections to isolate microsatellite loci were performed. Map layout from http://d-maps.com/carte. php?num_car=1521&lang=en.

(*Fernandez-Silva et al., 2013*). Reads were trimmed for adapters and quality using the FASTX-Toolkit. The software Newbler 2.3 (Roche, Basel, Switzerland) was used to perform the *de novo* assembly. The programs MSATCOMMANDER 0.8 (*Faircloth, 2008*) and SSRfinder were used to search for di-, tri-, tetra-, penta-, and hexa-nucleotide repetitions. Thirty-nine pairs of primers flanking the microsatellite regions were designed using Primer3 (http://bioinfo.ut.ee/primer3-0.4.0/) and primer characteristics were checked using OligoAnalyzer 3.1 (https://www.idtdna.com/calc/analyzer/). Forward primers were designed with a M13 tail at their 5′ end (TGT AAA ACG ACG GCC AGT) for dye labeled (6-FAM, VIC, NED, or PET) primers annealing to the replicated strand during PCR reactions (*Schuelke, 2000*).

A total of 47 specimens of *T. coccinea* and 48 *T. tagusensis* were amplified by Polymerase Chain Reactions (PCRs). PCRs were performed in 10 µl reactions including 0.2 µM of forward primer with M13 tail, 0.4 µM of labeled primer (M13 with VIC, NED, PET, or 6-FAM fluorescent dyes), 0.8 µM of reverse primer, 1U GoTaq (Promega, Fitchburg, WI, USA), 1× PCR Buffer (Promega), 0.20 mM dNTPs (Invitrogen, Carlsbad, CA, USA), between 1.5 and 2.5 mM $MgCl_2$ (Table 1), 10 µg BSA (Invitrogen), and 5–10 ng of DNA. Cycling conditions were: 95 °C for 3 min followed by 5 cycles at 95 °C, 30 s; 52–62 °C (Table 1), 30 s; 72 °C, 45 s; and 30 cycles at 92 °C, 30 s; 52–62 °C, 30 s; 72 °C, 55 s; with a final extension at 72 °C for 30 min (*Toonen, 1997*). Amplification was verified in 2% agarose gel. PCR products were pooled with GS600-LIZ size standard (Applied Biosystems, Waltham, MA, USA) and genotyped in the ABI 3500 genetic Analyzer (Applied Biosystems). Genotypes were determined using the program Geneious 7.1.9.

## Statistical analyses

Clonal structure of each species was assessed using the 'GenClone' on R 3.2.3 package (*R Core Team, 2015*). Samples with the same alleles at all loci (ramets) were assigned to the same multilocus genotype (MLG, or genets) and considered to be a product of asexual reproduction. To check if individuals with the same MLG were truly clones, the probability of finding identical MLGs, resulting from distinct sexual reproductive events ($P_{sex}$), was calculated following *Arnaud-Haond et al. (2007)*. When $P_{sex} < 0.001$, samples are considered ramets belonging to the same genet. In order to avoid the overestimation of genotype numbers due to scoring errors or somatic mutations (*Douhovnikoff & Dodd, 2003*), a second analysis calculating the genetic distance among all pairs of genets was performed. Based on the genetic distances, MLGs that differed at only one allele were assigned to the same multi-locus Lineage (MLL) (*Arnaud-Haond et al., 2007*). For the genetic diversity and population structure analyses, only unique MLLs were considered.

To assess the clonal structure of each population, two indexes were calculated as proposed by *Arnaud-Haond et al. (2007)*: (1) clonal richness, to evaluate the proportions of clones in each population ($R = G - 1/N - 1$), where $G$ represents distinct multilocus lineages (MLL) and $N$ is the total number of individuals sampled. The index ranges from zero (when all individuals are clones) to one (when all samples analyzed correspond to a different MLL); and (2) the genotypic evenness, to evaluate the equitability in the distribution of the MLL, calculated by the Simpson's complement evenness index ($V = (D - D_{min})/(D_{max} - D_{min})$), where $D$ represents the observed diversity, $D_{max}$ the value assumed if all genets have the same number of ramets, and $D_{min}$ the diversity value when all but one genet has one individual (*Hurlbert, 1971*). This index ranges from zero (when one genet dominates the population) to one (when genets each have the same number of ramets).

Quality control of loci followed *Selkoe & Toonen (2006)*. To assess each population's genetic diversity, the number of alleles (Na), observed (Ho) and expected heterozygosities (He) were calculated using the 'diveRsity' in R 3.2.3 package (*R Core Team, 2015*). Significant deviations from Hardy–Weinberg equilibrium (HWE) and linkage equilibrium were tested with the FSTAT program (*Goudet, 1995*). The occurrence of null alleles was investigated using the Micro-Checker program (*Van Oosterhout et al., 2004*). To measure

Peer*J*

**Table 1 Description of *Tubastraea coccinea* and *Tubastraea tagusensis* microsatellite loci with their respective GeneBank Accession number.** Forward primers include an M13 sequence (5′-TGTAAAACGACGGCCAGT-3′).

| Locus/ Accession number | Primer sequence | Repeat motif | Species | $T_A$ (°C)/ [ ] MgCl$_2$ (mM) | Range (bp) | TSB ($N = 23$-Tc/24-Tt) | | | | IBG ($N = 24$-Tc/24-Tt) | | | |
|---|---|---|---|---|---|---|---|---|---|---|---|---|---|
| | | | | | | Na | Ho | He | $F_{IS}$ | Na | Ho | He | $F_{IS}$ |
| Tco1/ KY198738 | F:TGTAAAACGACGGCCAGTACTTCGGTGATCGGACGAG-**PET** | (GTT)6 | *T. coccinea* | 56/2 | 567–600 | [a] | | | | | | | |
| | R: AGCACGGGTACTTGCTTTG | | *T. tagusensis* | 56/2 | | 2 | 0.12 | 0.18 | 0.00 | 1 | 0.00 | 0.00 | NA |
| Tco4/ KY198739 | F: TGTAAAACGACGGCCAGTGTGGAGAGTGAATAAGCTTGGG-**NED** | (TCA)4 | *T. coccinea* | 60/2 | 253–259 | 2 | 1.00 | 0.50 | −1.00 | 2 | 1.00 | 0.50 | −1.00 |
| | R: GCCTGATGGTTTCTTGAGGTC | | *T. tagusensis* | 58/2 | | 2 | 0.40 | 0.32 | −0.14 | 2 | 0.33 | 0.28 | 0.00 |
| Tco5/ KY198740 | F: TGTAAAACGACGGCCAGTTCAGGAGCCGATTAATACCTG-**6FAM** | (GAAA)5 | *T. coccinea* | 54/2 | 368–432 | 5 | 0.50 | 0.76 | 0.39 | 3 | 0.20 | 0.34 | 0.50 |
| | R: TGTGCAGTGAATGTGCTCAAG | | *T. tagusensis* | 54/2.5 | | 2 | 0.60 | 0.42 | −0,33 | 2 | 0.67 | 0.44 | −0.33 |
| Tco8/ KY198741 | F: TGTAAAACGACGGCCAGTGGTGCAGTGTAAATTGGTTCG-**PET** | (GGA)6 | *T. coccinea* | 54 /2 | 343–349 | 2 | 1.00 | 0.50 | −1.00 | 2 | 1.00 | 0.50 | −1.00 |
| | R: GACAAGTGGAAAGCGGACG | | *T. tagusensis* | 52/2 | | 2 | 1.00 | 0.50 | −1.00 | 2 | 1.00 | 0.50 | −1.00 |
| Tco9/ KY198742 | F: TGTAAAACGACGGCCAGTTTGACCACGTACTGCCAAG-**VIC** | (TA)10 | *T. coccinea* | 60/2 | 347–357 | [a] | | | | | | | |
| | R: TCTGTTCAGAGAGCTCCGC | | *T. tagusensis* | 60/2 | | 2 | 0.20 | 0.18 | 0.00 | 1 | 0.00 | 0.00 | NA |
| Tco29/ KY198743 | F: TGTAAAACGACGGCCAGTGTGCCCTAGGTCCATGGTTT-**VIC** | (ATA)20 | *T. coccinea* | 62/1.5 | 211–222 | 3 | 0.70 | 0.51 | −0.31 | 3 | 1.00 | 0.57 | −0.71 |
| | R: CCGGCTTCTATATAGGCTTCC | | *T. tagusensis* | 58/2 | | 3 | 0.20 | 0.46 | 0.64 | 1 | 0.00 | 0.00 | NA |
| Tco30/ KY198744 | F: TGTAAAACGACGGCCAGTGGGAATTCGGATGCAATTAT-**6FAM** | (ACAT)6 | *T. coccinea* | 60/1.5 | 252–264 | 3 | 1.00 | 0.61 | −0.63 | 3 | 1.00 | 0.58 | −−0.67 |
| | R: CTCTGTGGAATGAGCTGCAA | | *T. tagusensis* | 60/2.25 | | 2 | 1.00 | 0.50 | −1.00 | 2 | 1.00 | 0.50 | −1.00 |
| Tco32a/ KY198745 | F: TGTAAAACGACGGCCAGTGCGTGGTCTGGTCTTTTCAT-**6FAM** | (ATA)13 | *T. tagusensis* | 58/2 | 240–246 | 2 | 1.00 | 0.50 | −1.00 | 2 | 1.00 | 0.50 | −1.00 |
| | R: ACCCACTTTGAGGTGTTTGG | | | | | | | | | | | | |
| Tco32b/ KY198745 | [a] | | *T. tagusensis* | | 270–276 | 2 | 1.00 | 0.50 | −1.00 | 3 | 1.00 | 0.61 | −0,50 |
| Tco34/ KY198746 | F: TGTAAAACGACGGCCAGTGCGCCTACTACCACACGAAT-**PET** | (TTA)19 | *T. coccinea* | 58/2 | 189–217 | 2 | 0.38 | 0.31 | −0.20 | 2 | 0.17 | 0.15 | 0.00 |
| | R: TCCTTTCTACAGCGCACCTT | | *T. tagusensis* | 58/2 | | 3 | 0.80 | 0.58 | −0.28 | 3 | 1.00 | 0.61 | −0.50 |
| Tco36/ KY198747 | F: TGTAAAACGACGGCCAGTGCAATGACAACAGCCAGAAC-**VIC** | (ATA)15 | *T. coccinea* | 58/1.5 | 238–250 | [b] | | | | [b] | | | |
| | R: TTTCGTCTGCCACATTCTTG | | | | | | | | | | | | |
| Tco37/ KY198748 | F: TGTAAAACGACGGCCAGTAAACATTCGATTCCCACTCG-**NED** | (CTA)24 | *T. coccinea* | 62/1.5 | 242–263 | 4 | 1.00 | 0.74 | −0.32 | 2 | 1.00 | 0.50 | −1.00 |
| | R: ACCCGGCCACTAATATTTCC | | *T. tagusensis* | 62/1.5 | | 3 | 1.00 | 0.62 | −0.50 | 3 | 1.00 | 0.61 | −0.50 |
| Tco38/ KY198749 | F: TGTAAAACGACGGCCAGTTTTGAGTTTGAGTTTATTGACTCCTT-**NED** | (TACA)6 | *T. coccinea* | 58/1.5 | 227–235 | [b] | | | | [b] | | | |
| | R: GGAGTAAGCTTAGAGGGGTGCT | | | | | | | | | | | | |

**Notes.**

$T_A$, primer's annealing temperature; [ ], MgCl$_2$ concentration of magnesium chloride; $N$, number of individuals genotyped; Na, number of alleles; He, expected heterozygosity; Ho, observed heterozygosity; $F_{IS}$, inbreeding coefficient (negative values indicate an excess of heterozygotes).

[a] Loci with evidence of linkage disequilibrium.

[b] Loci with evidence of null alleles.

population structure two indexes were calculated using the programs Genetix (*Belkhir et al., 2004*) and GenoDive (*Meirmans & Van Tienderen, 2004*). (1) Wright's fixation index $F_{ST}$, ranging from zero, when different populations have identical alleles frequencies, to one, when each population has different fixed alleles (*Wright, 1965*). However, when applied to highly polymorphic markers, such as microsatellites, this index never reaches one and can underestimate genetic differentiation (*Hedrick, 1999*; *Meirmans & Hedrick, 2011*; *Bird et al., 2011*). The second measure, (2) Meirmans and Hedrick's differentiation index $G''_{ST}$, is a standardized measure rescaled from zero to one based on the maximum value of $G''_{ST}$ which simplifies interpretation of the degree of genetic differentiation among populations when using highly polymorphic microsatellite markers (*Meirmans & Hedrick, 2011*; *Bird et al., 2011*).

A Bayesian analysis was performed to estimate the number of genetic clusters in the dataset using STRUCTURE v. 2.3.4 software (*Pritchard, Stephens & Donnelly, 2000*) with the admixture ancestry model and correlated allele frequency. The analysis was performed with an initial burn-in of 500,000 cycles followed by 500,000 additional cycles and the number of clusters ($K$) tested varied from one to three with 15 iterations for each $K$-value. A higher range in the number of clusters ($K$ ranging from one to five) was also tested to verify possible substructure within the populations. The most likely $K$-value was estimated by estimating the "log probability of data" for each value of $K$ (mean LnP($K$)) (*Pritchard, Stephens & Donnelly, 2000*) using STRUCTURE HARVESTER (*Earl & Von Holdt, 2012*). The $\Delta K$ criterion, frequently used in population genetic studies, is applied for datasets with more than two populations and as one of the hypotheses here is that the two localities are one panmitic population, this criterion was not used in the present work (*Evanno, Regnaut & Goudet, 2005*).

## RESULTS

### Characterization of microsatellite markers

The two 454 runs resulted in a total of 329,832 reads with an average size of $\pm708.5$ bp. A total of 1,077 regions with 2–6 bp microsatellite repeats with at least four units were found. Among these regions, 39 were selected for primer design, based on the size and position of the repeat within the sequence, and the primer characteristics (e.g., lacking primer-dimer formation). Within these, 11 and 10 were successfully amplified and genotyped for *Tubastraea coccinea* and *T. tagusensis* respectively (Accession numbers: KY198738–KY198749). While two loci failed to amplify for *T. tagusensis* (Tco36 and Tco38), this species also exhibited two loci at a single locus with no evidence of linkage disequilibrium between them (Tco32a and Tco32b), so both were included in these analyses.

Evidence for null alleles for *T. coccinea* TSB population was observed in the same two loci (Tco36 and Tco38) that failed to amplify for *T. tagusensis*. Since both loci had only homozygote genotypes at the two analyzed localities, these loci were removed from the genetic diversity analyses. The loci Tco1 and Tco9 showed evidence of linkage disequilibrium with other loci and were also removed from the remaining analyses. The
**Table 2  Genetic diversity of *Tubastraea coccinea* and *T. tagusensis* in two localities on the Southwestern Atlantic Ocean, Todos os Santos Bay (TSB) and Ilha Grande Bay (IGB), Brazil.**

| Specie | Location | N | MLG | MLL | R | V | A | AR | Ap | Ho | He | $F_{IS}$ |
|---|---|---|---|---|---|---|---|---|---|---|---|---|
| *T. coccinea* | TSB | 23 | 13 | 13 | 0.55 | 0.845 | 21 | 2.74 | 4 | 0.80 | 0.56 | −0.380 |
| | IGB | 24 | 6 | 6 | 0.21 | 1.13e−16 | 17 | 2.18 | 0 | 0.77 | 0.45 | −0.651 |
| *T. tagusensis* | TSB | 24 | 7 | 5 | 0.17 | 0.54 | 25 | 1.98 | 4 | 0.67 | 0.43 | −0.468 |
| | IGB | 24 | 6 | 3 | 0.09 | 1.04e−16 | 22 | 1.87 | 1 | 0.64 | 0.37 | −0.615 |

Notes.

$N$, Number of individuals sampled; MLG, multilocus genotype; MLL, multilocus lineages; R, clonal richness; V, genotypic evenness; β, pareteo distribution; A, alleles number; AR, allele richness; Ap, number of private alleles; Ho, observed heterozigosities; He, expected heterozigosities; $F_{IS}$, inbreeding coefficient.

number of alleles per locus ranged from one to five in *T. coccinea* and one to four in *T. tagusensis*. Between localities, Ho ranged from 0.38 to 1 (TSB) and 0.17 to 1 (IGB) for *T. coccinea* and from 0.2 to 1 (TSB) and 0 to 1 (IGB) for *T. tagusensis*. He ranged from 0.31 to 0.76 (TSB) and 0.15 to 0.58 (IGB) for *T. coccinea* and from 0.18 to 0.62 (TSB) and 0 to 0.61 (IGB) for *T. tagusensis* (Table 1). In general, the observed heterozygosity was higher than expected for most loci in both populations of both species, with up to 100% of individuals being heterozygous at some loci (Table 1), although no significant deviation from HWE was observed.

## Clonality

$P_{sex}$ values observed were highly significant (<0.001) for all but two and seven individuals of *T. coccinea* and *T. tagusensis* respectively. Thus, these data do not support the hypothesis of several individuals with the same MLG having originated by chance from distinct sexual reproduction events. A high proportion of clones were observed at both localities for both species (Table 2). For *T. coccinea*, at TSB of the 23 colonies sampled 13 MLLs were found, while at IGB only six MLLs out of the 24 colonies sampled were found. *T. tagusensis* had five (at TSB) and three (at IGB) unique MLLs among the 24 sampled colonies at each locality (Table 2). Missing values were considered as different alleles by the program, and although only specimens with missing information at no more than one locus were kept, it is important to note that the final number of MLL might be overestimated slightly.

Clonal richness observed for *T. coccinea* indicates that IGB is mostly composed of clones ($R = 0.22$), with only six MLLs out of 24 individuals, while TSB has nearly half of the individuals comprised of clones (13 MLL in 23 individuals sampled; $R = 0.55$) (Table 2). In addition to the low MLL diversity at IGB, 19 individuals had the same predominant MLL, which was observed by the evenness indexes ($V = 1.13^{-16}$). Conversely, the TSB population of *T. coccinea* had more equally distributed MLLs, with the most common one being shared among only 4 individuals ($V = 0.85$). For *T. tagusensis*, both populations were composed mainly of clones, with very low clonal richness (IGB: $R = 0.09$; TSB: $R = 0.17$). Similarly to what was observed for *T. coccinea*, MLLs were more equally distributed at TSB, with 14 individuals belonging to the same MLL ($V = 0.54$), while in IGB the most common one was shared among 22 individuals ($V = -1.04^{p-16}$).

### Genetic diversity and population structure

Only unique MLLs were used to assess genetic diversity and population structure in each species. For both species, TSB had higher number of alleles, allelic richness and number of private alleles compared to IGB, with *T. coccinea* presenting the more accentuated differences (Table 2). There were no significant deficits of heterozygosity; both observed (Ho) and expected (He) heterozygosity were similar when comparing between localities for both *T. coccinea* (TSB: 0.80 and 0.56; IGB: 0.77 and 0.45) and *T. tagusensis* (TSB: 0.67 and 0.43; IGB: 0.64 and 0.37). The inbreeding coefficient ($F_{IS}$), although not significant, was negative for both localities and in both species, indicating an excess of heterozygotes (Table 2).

$F_{ST}$ and $G''_{ST}$ values were 0.06 ($p = 0.08$) and 0.13 ($p = 0.07$) for *T. coccinea* and indistinguishable from zero ($p = 0.69$ and $p = 0.69$) for *T. tagusensis*. The lack of significant population structure among the sampled localities indicates similar allele frequencies for both species across these sites. Although Bayesian analysis recovered two genetic clusters for *T. coccinea* for both ranges of $K$ tested, these groups are not a function of population structure between localities (Fig. 2), but instead, reflect the presence of population structure within each locality. Furthermore, there is no evidence of interbreeding between the two clusters, and the $F_{ST}$ values between these sites is likely a result of the strikingly different proportion of these two groups in each site. In contrast, no clustering was observed between or within localities for *T. tagusensis*, with the most likely $K$ value being one for both ranges of $K$ tested (Fig. 2).

## DISCUSSION

The novel microsatellite markers reported herein will enable further studies regarding the genetic diversity and population structure of *Tubastraea* spp. corals in the Atlantic and native ranges of these invasive populations. Using these microsatellites, this study shows that both invasive coral species (*T. coccinea* and *T. tagusensis*) have high proportions of clones at both localities on the Brazilian coast with identical multilocus lineages (MLLs) found in sites separated by more than 1,500 km. The results indicate that asexual reproduction dominates in the invasive range of *Tubastraea* spp. in the Southwestern Atlantic and despite the large distance between localities, no significant population structure could be found. In contrast, there are clear signs of population structure across this same region in an endemic spawning coral species (*Mussismilia hispida*, *Azevedo, 2015*).

Our results support previous work reporting reproduction via asexual larvae in *T. coccinea* (*Ayre & Resing, 1986*). Likewise, the high proportion of clones found at both sampled localities for *T. tagusensis* indicates likely reproduction by asexual larvae for this species also, a reproductive mode previously recorded for only three scleractinian species: *Pocillopora damicornis* (*Stoddart, 1983*), *Tubastraea diaphana* and *T. coccinea* (*Ayre & Resing, 1986*). Indeed, a study on the reproductive strategies of *T. coccinea* and *T. tagusensis* in the Southwestern Atlantic observed a small number of spermaries and the presence of embryos and planula at different times of the year, concluding that asexual reproduction could be important for both species (*De Paula, Pires & Creed, 2014*). For most corals,
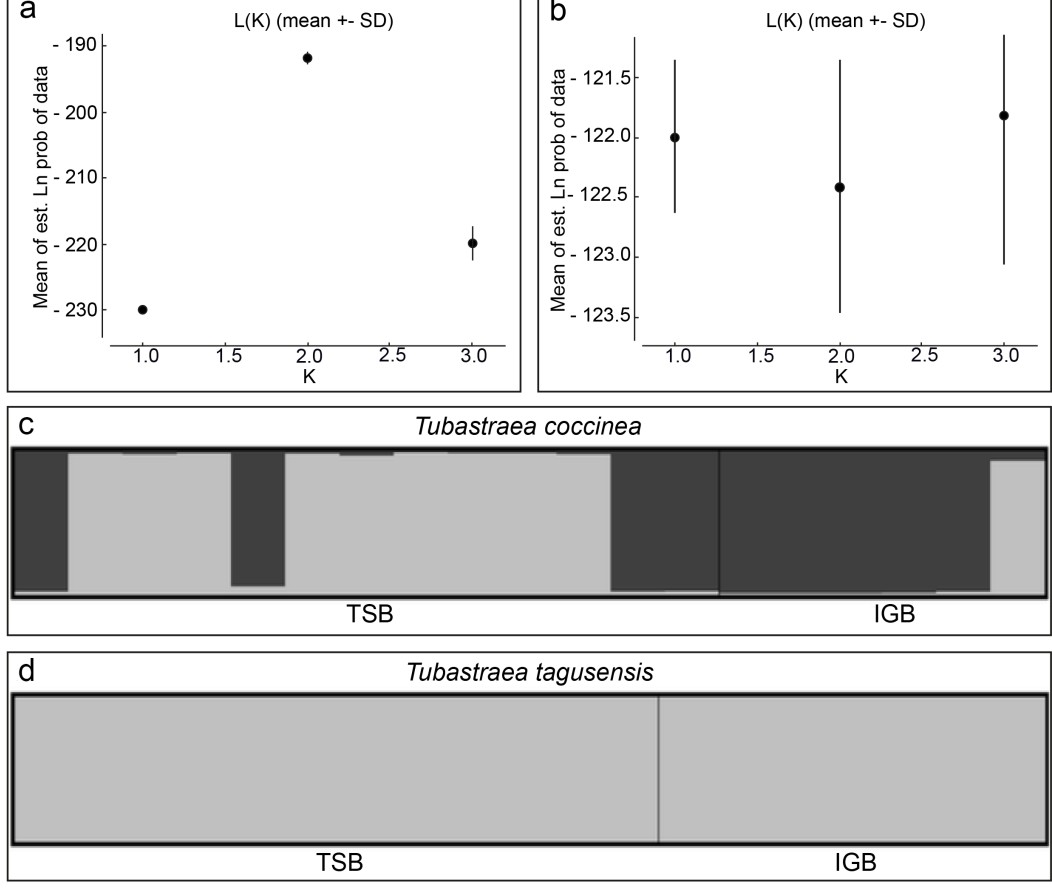

**Figure 2 Bayesian clustering analyses for *Tubastraea coccinea* and *T. tagusensis*.** (A) and (B) shows the most likely *K*-value estimated by the mean of estimated "log probability of data" for each value of *K* for *T. coccinea* (*K* = 2) and *T. tagusensis* (*K* = 1), respectively; (C) and (D) shows the genetic clusters, where each individual is represented by a vertical bar with different colors indicating the relative proportion of each genetic cluster. TSB, Todos os Santos Bay; IGB, Ilha Grande Bay.

clonality is a result of mechanical fragmentation due to physical disturbances (*Foster et al., 2013*; *Nakajima et al., 2015*). *T. coccinea* and *T. tagusensis*, however, are not prone to fragmentation, so the high number of clones observed for both species in this study seems more likely to result from asexually produced larvae. Nevertheless, it is desirable to confirm the production of asexual larvae for both *T. coccinea* and *T. tagusensis* by performing paternity studies in the future.

For invasive species, asexual reproduction can be crucial in the first stage of invasion, when sexual partners are scarce or absent, because it significantly enhances the chances of survival for the colonists (*Taylor & Hastings, 2005*). Successful invasions originating from a few clonal genotypes have been previously recorded for plants (*Ren, Zhang & Zhand, 2005*; *Liu et al., 2006*) and other cnidarians (*Reitzel et al., 2008*). Asexual reproduction is dominant in the invasive range and it may have contributed to the invasive success of *Tubastraea* in the Southwestern Atlantic, where the rocky shores provide a suitable habitat and release from enemies (Enemy Release Hypothesis, *Keane & Crawley, 2002*). At IGB

both studied coral species have high percentage of clones and an extremely low genotypic evenness, indicating that most colonies are clones belonging to the same genet. Sampling more areas surrounding each collection site is needed to thoroughly examine clonal diversity for these regions, but particularly in TSB where samples were more widely spaced, this observation supports the role of asexual reproduction in increasing local abundance. Gregarious settlement has been previously observed for both *T. coccinea* (*Glynn et al., 2008*; *De Paula, Pires & Creed, 2014*), and *T. tagusensis* (*De Paula, Pires & Creed, 2014*), although these studies did not determine if the aggregated larvae were sexually or asexually derived. It is noteworthy that *T. coccinea* has higher numbers of MLLs, clonal richness and genotypic evenness at TSB than at IGB, suggesting increased occurrence of sexual reproduction or a greater number of successful colonists at the former site. Rates of sexual and asexual reproduction can be highly variable among geographic regions in other corals (*Baums, Miller & Hellberg, 2006*; *Noreen, Harrison & Van Oppen, 2009*; *Combosch & Vollmer, 2011*; *Gorospe & Karl, 2013*), but it remains unknown what governs the difference in the proportion of sexual and asexual reproduction at different localities. Several factors can influence both genotypic and genetic diversity in invasive species, including the number of invasions, the genetic diversity of the source population(s) and a variety of biological factors, such as the main reproductive strategy adopted by the species (*Dlugosch & Parker, 2008*). Although sexual reproduction might also occur in *Tubastraea*, the results obtained for *T. coccinea* might be an effect of the occurrence of recent multiple introductions from different native populations (*Roman & Darling, 2007*). Another hypothesis would be the presence of cryptic species, which has been found in other scleractinian corals (*Pinzón & Weil, 2011*; *Warner, Van Oppen & Willis, 2015*; *Nakajima et al., 2017*). Morphological analyses combined with molecular data including native populations are necessary to corroborate this hypothesis.

A decrease in genetic diversity as a result of a small founding population has been previously recorded for several invasive populations (*Roman & Darling, 2007*; *Geller et al., 2008*; *Johnson & Woollacott, 2015*; *Wrange et al., 2016*; but see *Gaither et al., 2010*; *Gaither, Toonen & Bowen, 2012* for counter-examples). Here, we report excess of heterozygosity for both populations of both species and the presence of up to 100% heterozygous individuals at some loci (Table 1). High levels of heterozygosity can result from an isolate-breaking effect, when multiple introductions mix previously separated native populations (*Holland, 2000*; *Hamilton, 2010*). However, in this case, there is no evidence of mixing between the two genetic clusters (Fig. 2), indicating that they are not interbreeding. Thus, it seems more likely that TSB and IGB were colonized by different native populations followed by recent transport between localities without sufficient time for them to interbreed, although the possibility of cryptic species that are incapable of interbreeding should also be considered. If the first scenario of introduction by different native populations proves true, the high heterozygosity could be either a result of a founder effect in which the new area was, by chance, colonized by a higher number of heterozygote genotypes, or due to a higher fitness of the heterozygote genotypes, either of which could be propagated by asexual reproduction (*De Meeus & Balloux, 2005*). Alternatively, *Gaither, Toonen & Bowen (2012)* showed that introduced fishes in Hawai'i with a known history actually had higher and

more even genetic diversity than was observed in the native range, and such an effect could also explain the observed pattern here. In contrast to what is observed with *T. coccinea*, we recover only a single genetic cluster for *T. tagusensis* between both populations. This single cluster could result from either invasion of both localities from the same source population, or a secondary invasion along the Brazilian coast from the original locality being spread to another. Unlike *T. coccinea*, which is now considered cosmopolitan (*Cairns, 2000*), *T. tagusensis* has a restricted distribution (*Cairns, 1991*) and may have naturally low genetic diversity. The distinction between these species is reminiscent of the pattern reported by *Gaither, Bowen & Toonen (2013)* in which population structure of species in their native range predicts the diversity and rate of spread in the invasive range.

Considering that (i) both *T. coccinea* and *T. tagusensis* brood larvae competent for only ~18 days (in aquaria) that typically display gregarious settlement (*Glynn et al., 2008*; *De Paula, Pires & Creed, 2014*) and (ii) the absence of *Tubastraea* in extensive areas between the two localities, it is highly unlikely that they are connected through larval dispersal. On the other hand, oil platforms are known to be moved between these regions (*Sampaio et al., 2012*), and are considered the main vector for the introduction of *Tubastraea* into the southwestern Atlantic (*Castro & Pires, 2001*; *Creed et al., 2016*). Thus, our data showing a lack of structure between localities, and the occurrence of shared MLLs for each species among these distant sites, indicate that anthropogenic vectors, such as oil platforms, monobuoys, or other vessels have played an important role in dispersing these alien invasive species, and possibly assisting other species to spread along the coast (*Almeida et al., 2015*; *Creed et al., 2016*).

## CONCLUSIONS

Invasive *Tubastraea* spp. are spreading quickly throughout the Atlantic, in some areas covering up to 100% of the available surface (*Mantellato et al., 2011*) and outcompeting native and endemic species (*Mantellato et al., 2011*; *Santos, Ribeiro & Creed, 2013*; *Creed, 2006*). Despite this documented impact and concern, little is known about the genetic diversity and reproductive strategies of *Tubastraea* species globally. This study provides the first survey of genetic diversity and likely reproductive strategies along the southwestern Atlantic coast, demonstrating that asexual reproduction has an important role in the population dynamics of both *T. coccinea* and *T. tagusensis* and is probably a relevant feature leading to their invasive success. Results also indicate that there were likely at least two different populations of *T. coccinea* introduced into the southwestern Atlantic. A molecular systematic examination of the genus is highly recommended in order to check for the occurrence of cryptic species. Future studies should focus on the identification of potential source populations and the global phylogeography of *Tubastraea* with the goal of tracking and limiting future invasions, as well as the establishment of effective management and prevention strategies.

## ACKNOWLEDGEMENTS

We are grateful to Antonio Solé-Cava for helping editing the first set of Next-Generation Sequencing data and to Marcelo Mantellato and Projeto Coral-Sol for samples. We are also thankful to Diane Bailleul for all the support on the clonal analyses. This is Scientific Contribution No. 30 of the Projeto Coral-Sol.

### Funding

This work was supported by Coordenação de Aperfeiçoamento de Pessoal de Nível Superior (Joel C. Creed, Ciências do Mar 1137/2010); Fundação Carlos Chagas Filho de Amparo à Pesquisa do Estado do Rio de Janeiro (Katia Cristina Cruz Capel, Joel C. Creed and Carla Zilberberg, FAPERJ-E-26/010.003031/2014 PensaRio; Joel C. Creed E26/201.286/2014); Conselho Nacional de Desenvolvimento Científico e Tecnológico (Joel C. Creed, CNPq-305330/2010-1) and Fundação de Amparo à Pesquisa do Estado de São Paulo (Marcelo V. Kitahara, FAPESP 2014/01332-0); and Award NSF-OA#14-16889 (National Science Foundation). The funders had no role in study design, data collection and analysis, decision to publish, or preparation of the manuscript.

### Grant Disclosures

The following grant information was disclosed by the authors:
Coordenação de Aperfeiçoamento de Pessoal de Nível Superior: 1137/2010.
Fundação Carlos Chagas Filho de Amparo à Pesquisa do Estado do Rio de Janeiro: FAPERJ-E-26/010.003031/2014, E26/201.286/2014.
Conselho Nacional de Desenvolvimento Científico e Tecnológico: CNPq-305330/2010-1.
Fundação de Amparo à Pesquisa do Estado de São Paulo: FAPESP 2014/01332-0.
National Science Foundation: NSF-OA#14-16889.

### Competing Interests

Robert J. Toonen is an Academic Editor for PeerJ.

### Author Contributions

- Katia Cristina Cruz Capel conceived and designed the experiments, performed the experiments, analyzed the data, wrote the paper, prepared figures and/or tables, reviewed drafts of the paper.
- Robert J. Toonen wrote the paper, reviewed drafts of the paper.
- Caio T.C.C. Rachid analyzed the data, reviewed drafts of the paper.
- Joel C. Creed contributed reagents/materials/analysis tools, wrote the paper, reviewed drafts of the paper.
- Marcelo V. Kitahara contributed reagents/materials/analysis tools, wrote the paper, reviewed drafts of the paper.
- Zac Forsman wrote the paper, reviewed drafts of the paper.
- Carla Zilberberg conceived and designed the experiments, contributed reagents/materials/analysis tools, wrote the paper, reviewed drafts of the paper.
## Field Study Permissions

The following information was supplied relating to field study approvals (i.e., approving body and any reference numbers):

The samples used in the study were collected under the permit No 003/2014 from Instituto Brasileiro do Meio Ambiente e dos Recursos Naturais Renováveis – Ministério do Meio Ambiente, Brazil.

## DNA Deposition

The following information was supplied regarding the deposition of DNA sequences:

All microsatellite developed will be available via GenBank accession numbers KY198738 to KY198749. Sequences were uploaded as File S1.

## Data Availability

The raw data has been uploaded as a Supplemental File.

## Supplemental Information

Supplemental information for this article can be found online at http://dx.doi.org/10.7717/peerj.3873#supplemental-information.

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
