# Peer review of "Clone wars: asexual reproduction dominates in the invasive range of Tubastraea spp. (Anthozoa: Scleractinia) in the South-Atlantic Ocean"

_PeerJ, doi:10.7717/peerj.3873_

## Round 0.1 · original submission · Major Revisions

I have heard back from two reviewers, both of whom had generally positive things to say about your manuscript. However, both have also offered many constructive comments, and it will take some time to respond to all of their remarks. Thus, my decision is "major revisions" are needed.

Reviewer 1 ·

Basic reporting

Please see General comments below.

Experimental design

Please see General comments below.

Validity of the findings

Please see General comments below.

Additional comments

In this paper, the authors examined the genetic population structure of invasive Tubastraea species in the South-Atlantic Ocean by using microsatellite markers which have been newly developed by their study. Their analyses seem good and the interpretation of the results is also valid. Thus, I recommend the publication in PeerJ. I show some comments below.

When reading the manuscript first, it seemed difficult to understand the present distribution patterns of Tubastraea species used in their study. I recommend that the authors add the information in the map (Figure 1).

In the result, the lower genetic diversity around IGB for both species seems intriguing. Is IGB the marginal area for their distributions? If some environmental data such as temperatures are available, it would be informative to interpret the results (for example, please see the paper below).

Reference:
Dimond, J. L., Kerwin, A. H., Rotjan, R., Sharp, K., Stewart, F. J., & Thornhill, D. J. (2013). A simple temperature-based model predicts the upper latitudinal limit of the temperate coral Astrangia poculata. Coral Reefs, 32(2), 401-409.

L323: About the possibility of cryptic species, more descriptions may be helpful for the discussion. Please refer to the paper below too. This paper also discusses clonal diversity and the existence of cryptic species in coral shown by microsatellite analysis.

Reference:
Nakajima, Y., Nishikawa, A., Iguchi, A., Nagata, T., Uyeno, D., Sakai, K., & Mitarai, S. (2017). Elucidating the multiple genetic lineages and population genetic structure of the brooding coral Seriatopora (Scleractinia: Pocilloporidae) in the Ryukyu Archipelago. Coral Reefs, 36(2), 415-426.

Other comments:
L130: the authors should add the citation for FASTX-Toolkit.
L139 and other lines: "uL" -> "micro-L"
L139: "10ul" -> add space
L142: "MgCl2" -> "2" should be subscript.
L144 and other lines: "52°C–62°C" -> 52–62°C
L199: "hypothesis" -> "hypotheses"
L256: "were similar" -> "were higher"?
L305: "to a thoroughly examine" -> "to thoroughly examine"
Table 2: "," -> "."

Reviewer 2 ·

Basic reporting

This manuscript is well written. However, there are a few things that should be revised or clarified before this manuscript is published.

Line 26: azooxanthellate
Line 87: monobuoys
Line 88: citation please
Lin 165: MLL, term needs to be specific and identical in whole manuscript
Line 298-299: this sentence is not appropriate here. It is better to cite cases from animals instead of plants. There are some other examples from animals such as cnidarians that have been reported.

Table 2. He values of T. tagusensis should be with dot.

Experimental design

There are 4 collection sites (Buzios Island, monobuoy, Todos-os-Santos Bay (northeastern Brazil), and Ilha Grade Bay (southeastern Brazil)) as described in the Materials and Methods. Why only the result of the genetic connectivity between two localities have been shown in the study?
What are the genetic structures of samples from Buzios Island and monobuoy at Sao Sebastiao channel?

Line 193: The initial burn-in is 500,000 cycles followed by 500,000 additional cycles. How the number of burn-in cycles and run length have been determined?
Line 194: why the test of K varied from 1 to 3 only?

Validity of the findings

Since the asexual reproduction is the nature of these species which could cause the low genetic diversity overall. I would suggest the authors to increase the sample size to get more understanding regarding the origin of the invasive Tubastraea which is more relevant to the goal here. For example, if these species were transferred by the vessels, is it possible to collect some samples from the harbors and compare the data with those from the other areas, so we may see a picture of how these species migrate or spread?

Since the Tubastraea was first reported in the Caribbean and is now in Gulf of Maxico and Brazilian coast. It would be great if the authors perform the phylogeographic analyses to show the relationship of these species in the Brazilan coast and other places. By doing this, this will be clear to see how the invasion affect the local population.

The Tubastraea coccinea has been shown high dispersal potential (at least on the scale of tens of kilometers) (Mizrahi et al. 2014). Although this impact might be ruled out from the sites between TSB and IGB, can it affect the populations from the locations nearby, such as the localities in the south?
Also, I would suggest the authors to add an inset in Figure 1 to illustrate the collection sites in the south, so it could be helpful to understand the geographic distance among these sites.

The results also indicated at least two different populations of T. coccinea have been introduced to the southwestern Atlantic. Is it possible to detect the gene flow? By doing this, it will be helpful to understand where is this species from.

I also suggest authors to describe the asexual reproduction mode (s) of Tubastraea in the Introduction. Do T. coccinea and T. tagusensis have the same asexual reproduction strategies? Are these modes important for the expansion of these species?

Additional comments

The study aimed to identify the origin of the invasive Tubastraea spp. along the Brasilian coast by using microsatellite approach. The results showed the high proportions of clones at two analysed localities on Brazilian coast, indicating the dominate asexual reproduction which leads the conclusion that no significant population structures have been observed. The authors also concluded that the vector transport is the cause for spreading the species along the coast.
I appreciate that the authors share these dataset and analyses which will be important for the marine conservation management in the future.

---

## Round 0.2 · Minor Revisions

I have heard back from one reviewer, who has only a few minor comments left for you to address. Hence, my decision is "minor revision" are needed.

Reviewer 2 ·

Basic reporting

1. The authors should check the English before the publication.

Line 26: azooxanthellate

2. From the response, it seems the goal of this study is not to identify the origin of the invasive population as what is said in the manuscript. To make less confusing, I suggest the authors should remove this sentence in the Introduction "and the origins of the invasive populations remains a mystery", or modify it to something like "dispersal of the invasive population remains unclear".

Experimental design

no comment

Validity of the findings

no comment

Additional comments

no comment

---

## Round 0.3 · accepted · Accept

Thank you very much for your revisions - I look forward to seeing the published version of your work!